# CFD Simulation Study on the Cooling Characteristics of Shading and Natural Ventilation in Greenhouse of a Botanical Garden in Shanghai

Jianhong Shi [1], Haidong Wang [1,*] and Jianan Wang [2]

1    School of Environment and Architecture, University of Shanghai for Science and Technology, Shanghai 200093, China
2    Mechanical Intelligent System Commissioning and Test Center, Shanghai Installation Engineering Group Co. Ltd., Shanghai 200080, China
*    Correspondence: whd@usst.edu.cn

**Abstract:** Botanical garden greenhouses typically use solar radiation as an important heat source and meanwhile provide light for plants to survive. However, in the summertime, when the solar radiation is too strong, overheating will occur in the greenhouse and natural ventilation assisted with shading is used to cool it down. The modulation strategy of shading is very important not only to indoor temperature but also to the growth of plants. In order to determine the control strategy of the shading area in the design and installation stage, a CFD model of an exhibition greenhouse in Shanghai is established. During summer conditions, under the worst-case scenario of a windless day, the minimum shading area needed under different outdoor comprehensive temperatures is studied, and the correlation curve is fitted to guide the control of the shading to maintain appropriate thermal conditions. The decrease in indoor temperature under different shading areas is also explored when the outdoor comprehensive temperature is 34 °C. The annual carbon emission reduction of the greenhouse is about 500 t $CO_2$, by adopting shading and natural ventilation. This study provides a reference value for shading control and energy saving and emission reduction of a botanical garden greenhouse.

**Keywords:** botanical greenhouse; shading area; carbon emissions; CFD simulation; natural ventilation



## 1. Introduction

The botanical garden is not only a place to preserve endangered plants, but also an important base for scientific research and education. China has been experiencing a boom in the construction of botanical gardens, with around 170 completed botanical gardens all over the country [1]. The greenhouses of botanical gardens have very strict requirements for the environment to ensure the survival and normal growth of plants in greenhouses. Therefore, the indoor thermal environment of greenhouses needs to be strictly controlled. In contrast to conventional buildings, greenhouses generally have large transparent enclosures to ensure the natural light needed for plants to grow and to take full advantage of the heat from solar radiation to raise the temperature of the room. Meanwhile, when the outdoor comprehensive temperature is high, the excessive solar radiation heat must be controlled to ensure that the growing environment is not overheated. Natural ventilation is an effective way to eliminate overheating in greenhouses [2–5], but with the increase in outdoor temperature and solar radiation intensity, when the natural ventilation cannot eliminate the excessive heat gain from solar radiation, shading is the most commonly used cooling method in a greenhouse to further cool down the greenhouse [6–8]. Considering that plants usually need sunlight for photosynthesis, shading is usually the second option for cooling a greenhouse, except for some shade-loving plants, and the first option when the greenhouse overheat occurs is to employ opening on the envelope to bring outdoor air

in. How to adjust the shading under different outdoor environments (mainly outdoor air temperature and solar radiation intensity) is the key to the greenhouse environment.

The research on greenhouse ventilation and shading can be carried out by theoretical calculation and experimental measurement, and an experiment consumes extensive time, manpower, and material resources, and is not easy to be carried out in the stage of project design and demonstration. A computational fluid dynamics (CFD) simulation can effectively avoid the limitations of theoretical calculation methods in complex practical engineering and is usually employed in the design stage for conceptual study. In recent years, with the development of CFD technology, it is widely used in greenhouse internal environment simulation [9–14]. Gurpreet Singh et al. [15] established a mathematical model of the greenhouse microclimate, including four component equations of cover, inside air, canopy surface, and bare soil surface, and successfully predicted the change in indoor air temperature and humidity. Cheng et al. [16] introduced global variables of different categories such as indoor and outdoor temperature and humidity and proposed a global variable prediction model for the greenhouse, which provided a reference value for the follow-up study of greenhouse environmental control. Based on the principle of energy balance and CFD simulation, Piscia et al. [17] put forward a coupling method to study the environment and climate of the greenhouse at night, which provides a reference for greenhouse management. Using the LES model, Chu et al. [18] simulated greenhouses with plants to study the effects of plants on their natural ventilation, and the results showed that plants significantly reduced the ventilation rate. By using CFD technology, He et al. [19] studied the influence of the size of the ventilation opening on the cooling effect of the greenhouse. The results showed that the vent is an important factor affecting the temperature of the greenhouse, and the opening of the back wall has a significant effect on the cooling effect. Liang et al. [20] established a three-dimensional CFD model of natural ventilation in a greenhouse in Lhasa, Tibet, and studied the influencing factors of ventilation rate in the greenhouse with strong solar radiation, which provided a reference for further study of the ventilation characteristics of the greenhouse.

The above CFD simulation studies mainly focused on the relationship between greenhouses and natural ventilation, while the relationship between greenhouses and shading is not sufficiently studied. Therefore, in order to quantitatively evaluate the temperature regulation of shading and explore the relationship between shading area and greenhouse temperature, this study uses an exhibition greenhouse in Shanghai to mainly study the shading control strategy. The indoor temperature field under different outdoor temperatures and different shading areas was studied by using the CFD simulation method, and the required minimum shading area under different outdoor temperatures was explored. The fitting curve between outdoor temperature and required shading area was correlated. This provides the greenhouse shading control strategy for similar types of exhibition greenhouses. In addition, this paper further calculates the greenhouse carbon emissions reduction by adopting these passive measures.

The research object of this study is a greenhouse exhibition hall of a botanical garden in Shanghai. The total building floor area is close to 38,000 m$^2$. The Botanical Exhibition Park mainly includes one visitor service center and three exhibition greenhouses, which are the Meaty World Pavilion, the Tropical Rainforest Pavilion, and the Garden of Clouds. The layout of the greenhouse is shown in Figure 1.

In this paper, the largest and the most complex venue, the Garden of Clouds, is simulated. The whole building is about 200 m long, 70 m wide, and 18 m high. The shape of the greenhouse is irregular, and the side wall is mainly a transparent glass curtain wall. It is a typical large-space building with a large-area transparent enclosure. The floor plan of the greenhouse is shown in Figure 2. The openings in the side wall of the greenhouse are mainly located 2.5 m and below and are uniformly distributed around the greenhouse. The opening area is about 380 m$^2$. In addition, there are groups of waist-high windows 0.6 m high in the glass curtain wall at 1 m above the ground. The roof openings are circular windows of various sizes, with a total opening area of about 10,800 m$^2$, accounting for

about 90% of the roof area. In the worst-case scenario, the outdoor wind velocity is 0 and the natural ventilation by opening these windows is mainly driven by the stack effect. In the simulation study, only the buoyancy-driven natural ventilation is considered.

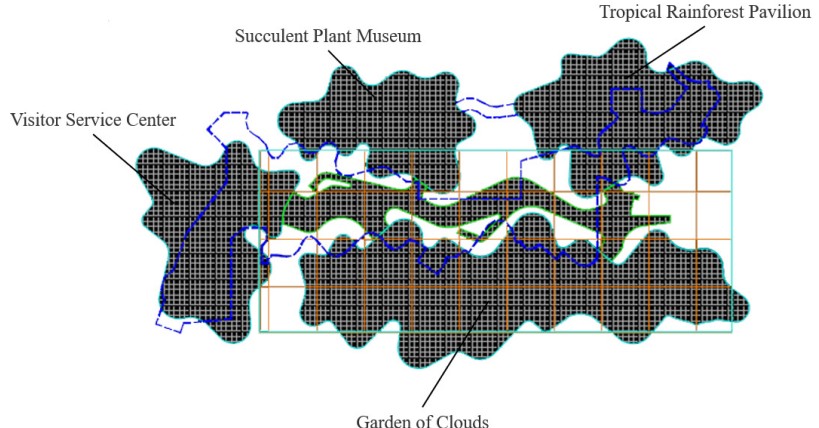

**Figure 1.** Top view of the botanical garden.

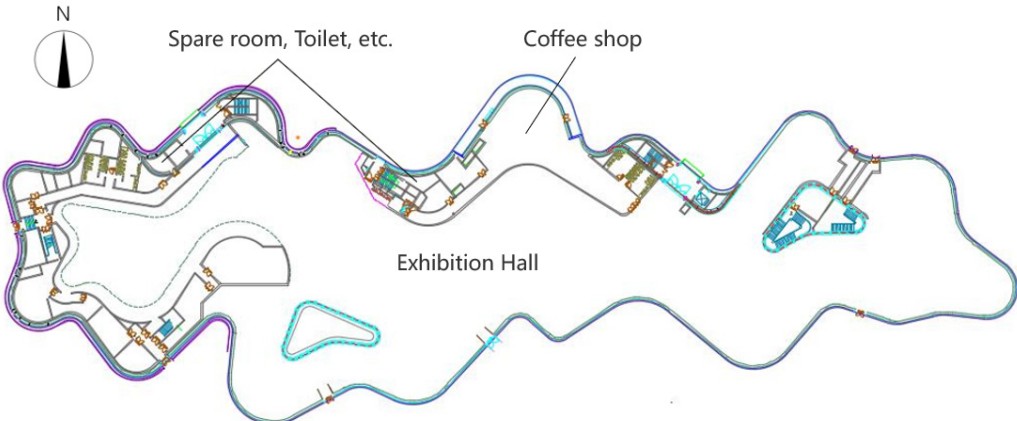

**Figure 2.** Floor plan of the greenhouse.

The roof openings are shown in Figure 3. Since the solar radiation intensity usually reaches its peak in the afternoon of the day, shading is placed on the west curtain wall of the greenhouse (the left area in Figure 2) and on the roof windows inside the building attached to the corresponding envelope. Interior shading in this facility significantly reduces the solar heat gain in the plant level, however, the heat accumulates near the shading instead of blocked out.

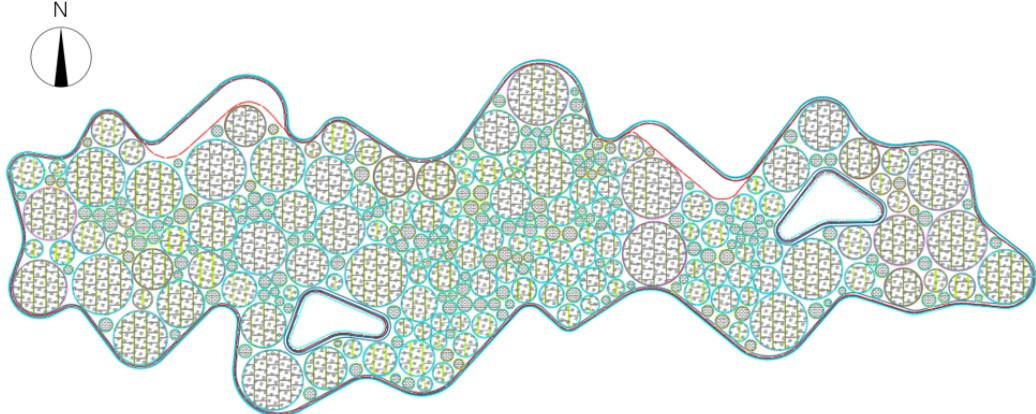

**Figure 3.** Top view of the greenhouse roof.

## 2. Methods

### 2.1. Heat Transfer Process of Greenhouse Transparent Envelope

The solar radiation mainly affects the greenhouse interior temperature through two aspects: on the one hand, because part of the solar radiation irradiates into the room through the transparent envelope, it will heat the interior surface through radiative heat transfer and exchange heat with the indoor air through convection; on the other hand, the absorption in the enclosure of the greenhouse will leave some heat in the glass, and lead to the increase in the temperature of the envelope to intensify the heat exchange with the indoor air by convection. The specific heat transfer process with and without shading is shown in Figure 4.

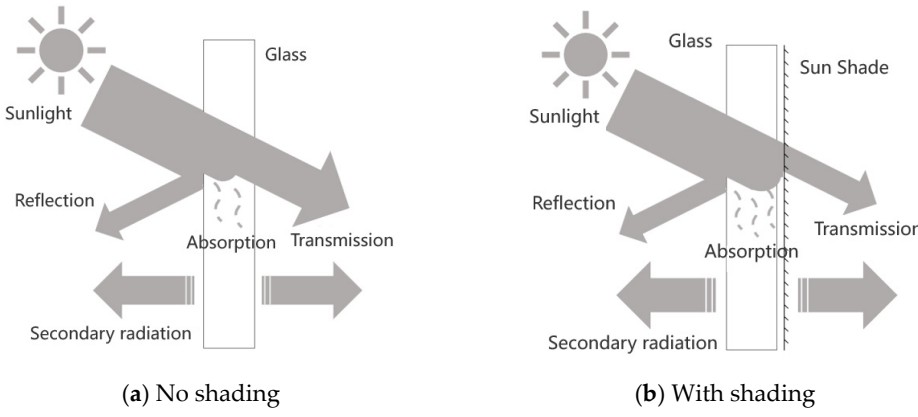

(**a**) No shading          (**b**) With shading

**Figure 4.** Principle of the heat transfer process of transparent envelope.

### 2.2. General Governing Equations

In the numerical simulation of the flow and heat transfer process in the indoor environment, the air is generally treated as an incompressible fluid. The general form of the fluid governing equation is Equation (1).

$$\frac{\partial}{\partial t}(\rho\phi) + div(\rho U\phi) = div(\Gamma_\phi grad\phi) + S_\phi \tag{1}$$

In the equation, $U$ is the velocity field, $\phi$ is the general variable, which represents the velocity vector of $u$, $v$, $w$, or air temperature $T$, and $S_\phi$ is the general source term. When using the computational fluid dynamics method for numerical simulation, the above-mentioned continuous partial differential equations are discretized into algebraic equations for numerical solutions in engineering applications. For turbulent flows, the Reynolds time-averaged method (RANS) is usually used to simplify the turbulence. In this study, the standard k-$\varepsilon$ model, which has been extensively validated in the previous study, is used to simulate the turbulent flow. For the near-wall region, the standard wall function is adopted, and the SIMPLE algorithm is adopted for the pressure–velocity coupling. Based on the Stephen–Boltzmann law, an IMMERSOL model in PHOENICS is used to calculate the radiant heat transfer between the solid surfaces. In this model, the formula for calculating the thermal conductivity is Equations (2)–(4).

$$\lambda = \frac{4\sigma T^3}{0.75(\varepsilon_1 + s_1 + \varepsilon_2 + s_2) + \frac{1}{W_{gap}}} \tag{2}$$

$$W_{gap} = 2(L'^2 + 2L)^{\frac{1}{2}} \tag{3}$$

$$\frac{d^2 L}{dy^2} = -1 \tag{4}$$

In the equations, $\lambda$ is thermal conductivity, W/(m·K), $\sigma$ is Stephen–Boltzmann constant, $\sigma = 5.67 \times 10^{-8}$ W/(m²·K⁴), T is temperature, K, $\varepsilon_1$ and $\varepsilon_2$ are surface radiation coefficients of solid and fluid, $s_1$ and $s_2$ are scattering coefficients of solid and fluid, respectively, and y is the distance between any point and wall.

*2.3. Modeling and Mesh Generation*

Commercial CFD software PHOENICS 2019 is used to build the model according to the actual size of the exhibition greenhouse. The calculation domain of the model is 200.5 m long, 68.9 m wide, and 18.0 m high. The left area of the greenhouse is the storage room and the spare machine room, while the upper area is mainly the tool room, the service room, and the machine room, leaving the rest of the region as the planting area. The summer cooling of the greenhouse venue mainly relies on natural ventilation and shading, and the plant area of the venue is not equipped with an air conditioning system. This paper mainly studies the minimum shading area needed under different outdoor comprehensive temperatures. For the plants in this exhibition greenhouse, the optimal growth temperature is 32–37 °C, and the indoor temperature should not exceed 40 °C. Considering the influence of solar radiation and other factors on greenhouse plants, the ratio of the shading area is calculated as less than 50% in the building design, so the shading area is 0–50% of the total envelope, and 5% is taken as the interval to represent different working conditions.

For the convective transfer of the exterior surface of the envelope, in addition to heat exchange with outdoor air, it is also affected by solar radiation, direct solar radiation, sky scattering radiation, ground reflection radiation, and long-wave radiation from the ground. In order to make the result more accurate, the convective heat transfer between the exterior surface of the envelope and the outdoor air and the effect of solar radiation heat are combined into one meteorological parameter, that is, the outdoor air comprehensive temperature [21]. The outdoor air comprehensive temperature is defined in Equation (5).

$$t_z = t_w + \frac{\rho_s I}{\alpha_e} \tag{5}$$

In the formula, $t_z$ is the outdoor comprehensive temperature, °C, $t_w$ is the outdoor air temperature, °C, $\rho_s$ is the solar radiation absorption coefficient, $I$ is solar irradiance, W/m², and $\alpha_e$ is the external surface heat transfer coefficient, W/(m²·K).

The outdoor comprehensive temperature of Shanghai in the cooling design day is 34 °C. Different outdoor comprehensive temperature working conditions are specified to 2 °C intervals. There are five working conditions, so the outdoor comprehensive temperature is between 26 °C and 34 °C. The top view of the greenhouse model is shown in Figure 5.

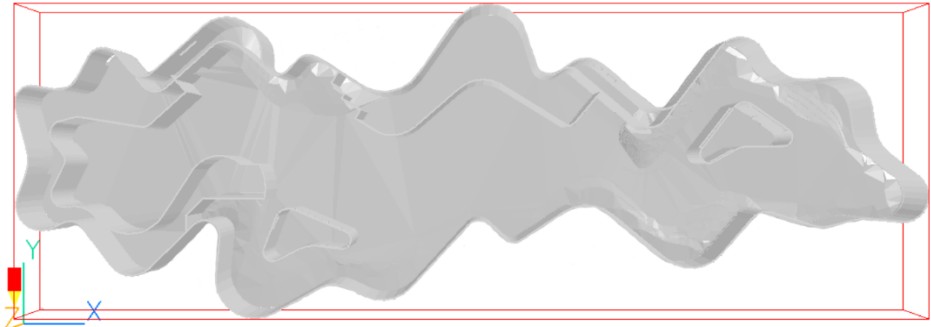

**Figure 5.** Top view of greenhouse model (hidden roof).

The solar radiation intensity involved in the study was calculated according to the solar radiation intensity of Shanghai at 2:00 p.m. on the calculation day. As the building adopts natural ventilation, window openings in both the side wall and roof are specified as openings of the pressure boundary. The effects of gravity and buoyancy are based

on the Boussinesq hypothesis, with the gravitational acceleration set to 9.81 m/s$^2$. The convergence criterion of the simulation is $10^{-3}$, and the number of iterations is about 1000. The relevant thermophysical parameters are shown in Table 1.

**Table 1.** Related thermophysical parameters.

| Name | Parameter | Value | Unit |
|---|---|---|---|
| Solar radiation | Solar radiation intensity | 836 | W/m$^2$ |
| Air | Density | 1.15 | kg/m$^3$ |
| | Thermal conductivity | 0.0267 | W/(m·K) |
| | Specific heat capacity | 1.005 | kJ/(kg·K) |
| Curtain wall | Density | 2500 | kg/m$^3$ |
| | Heat transfer coefficient | 1.8 | W/(m$^2$·K) |
| | Shading coefficient | 0.33 | — |

### 2.4. Verification of Grid-Independent Solutions

A structured grid was used to discretize the model, and 1.5 million (330 × 134 × 34), 3.36 million (478 × 138 × 51), and 6.62 million (564 × 178 × 66) meshes were selected to check the grid-independent solutions. The greenhouse model was meshed as Figure 6. In order to check the closeness of the mean air temperature prediction results of different grid resolutions at different heights, the temperature results of 10 different locations as illustrated in Figure 7 was sampled to reflect the overall thermal condition of the corresponding height level for validation. The validation result of the grid-independent solution is shown in Figure 8. The calculation results of different grid numbers are close to each other, and the maximum error is 0.9%. Therefore, in order to save calculation time, 1.5 million grid numbers are selected for further simulation study.

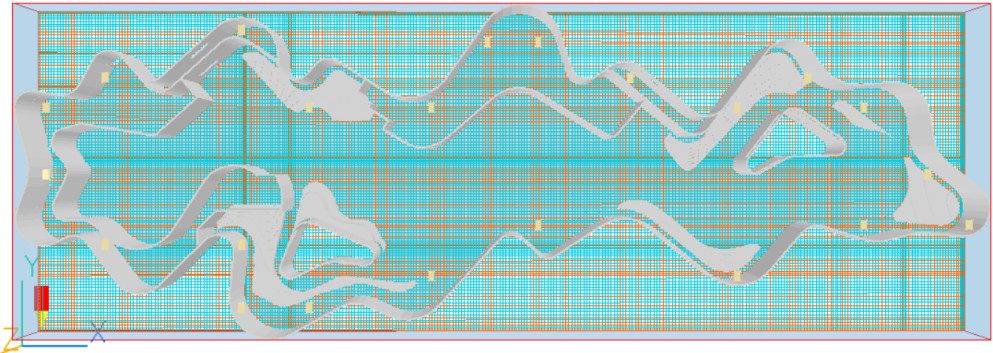

**Figure 6.** Top view of the computational grid.

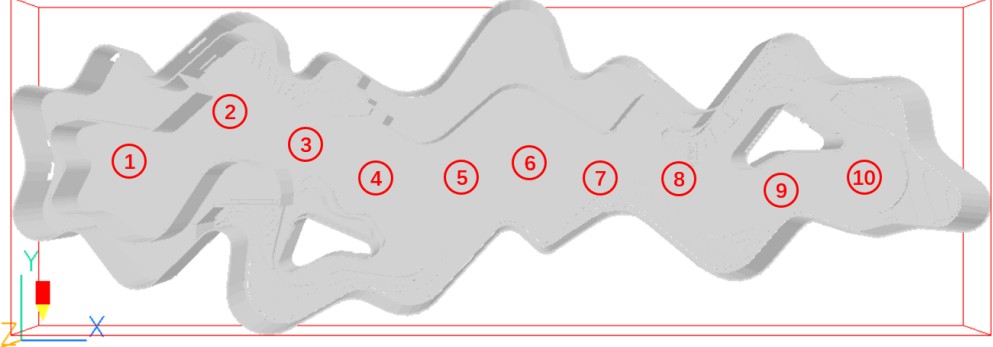

**Figure 7.** Distribution of the temperature sampling poles.

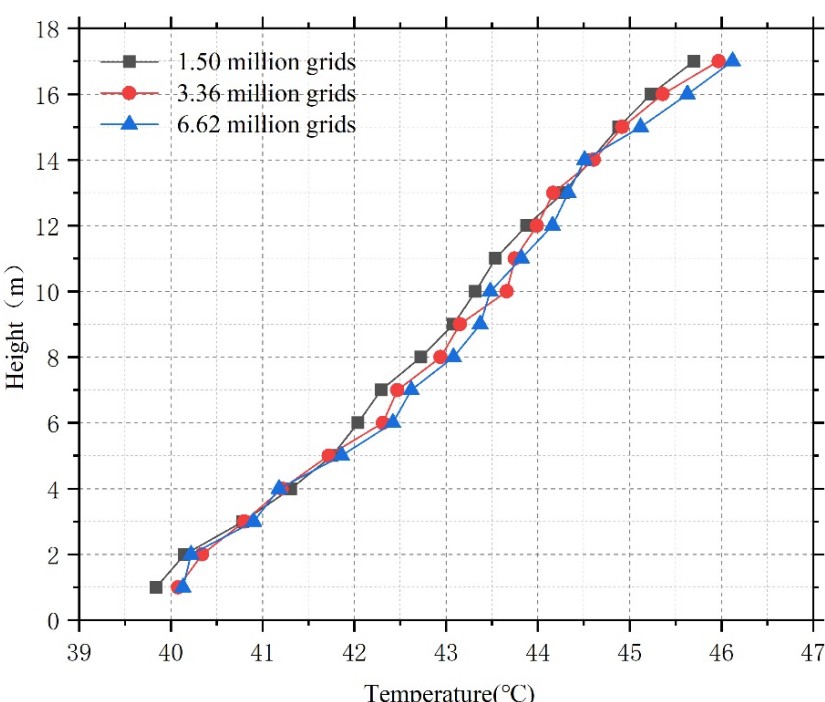

**Figure 8.** Average temperature distribution predicted by different grid numbers.

## 3. Results and Discussion

### 3.1. Indoor Temperature Control Results

Based on the design day meteorological temperature of Shanghai in summer, the outdoor comprehensive temperature of 34 °C at 2:00 PM (corresponding to the peak solar radiation) was used to simulate the greenhouse without shading. The results of the air temperature distribution inside the greenhouse are shown in Figures 9 and 10. It can be seen that the left side of the greenhouse has a higher temperature than the rest of the locations. This is mainly because the left side of the greenhouse is the area of the mechanical room, which has much less openings than the right side. Therefore, natural ventilation is not enough to cool down this area. There is obvious thermal stratification in the vertical direction according to Figure 10, and the temperature difference between the top and the bottom is about 4–5 °C.

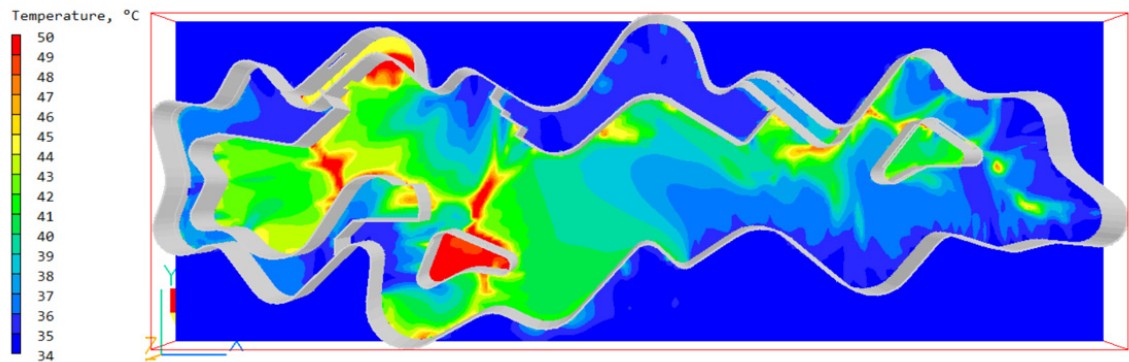

**Figure 9.** Contour of horizontal temperature distribution at 2 m high.

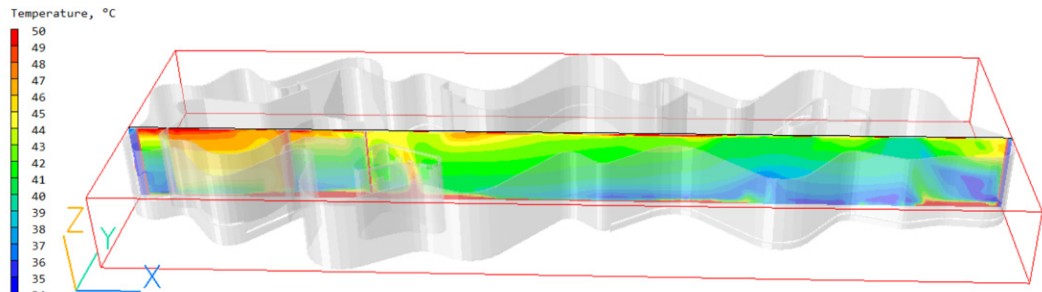

**Figure 10.** The contour of vertical temperature distribution 30 m away from the front.

In order to reflect the average temperature of a certain level, 10 vertical measuring poles were evenly placed in the greenhouse as illustrated in Figure 7, and one measuring point was arranged every 1 m on each pole to sample the temperature data. The average temperature of all measuring points is treated as the average indoor temperature in the following study. The vertical temperature distribution of each pole is shown in Figure 11.

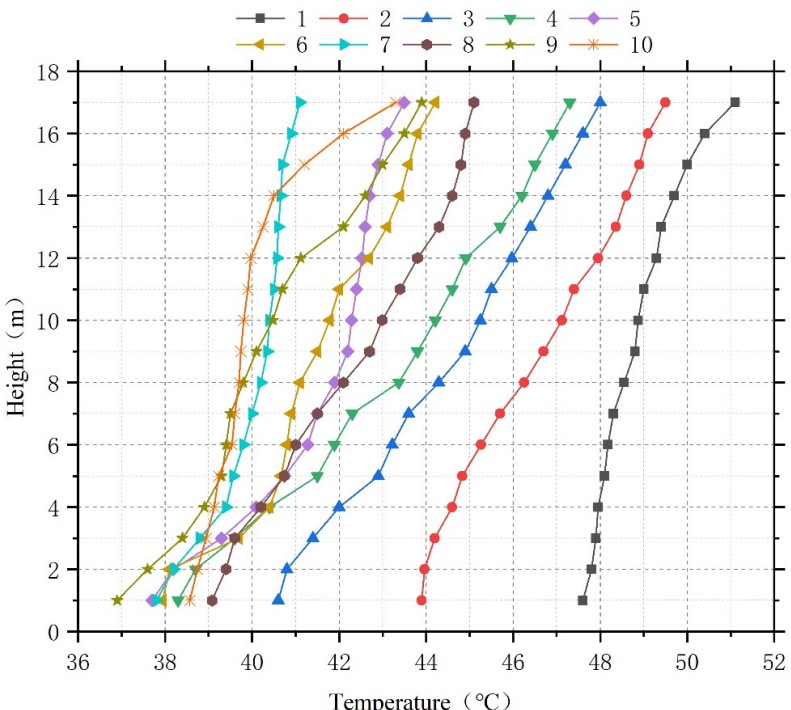

**Figure 11.** Temperature distribution of each measuring line.

The maximum temperature for the plant survival and growth is 40 °C, but because of the obvious thermal stratification and that the plant height is lower than that of the greenhouse, we take the average temperature of the greenhouse at 40 °C as the final cooling target. As can be seen from the above figure, even if all the vents are opened, the natural ventilation system cannot cool the greenhouse down to this temperature. Therefore, shading measures need to be taken to further reduce the room temperature. Due to the relatively poor natural ventilation on the left, resulting in a relatively high temperature in the area, shading was first placed on the left part of the roof. The indoor air temperature of about 40 °C can be obtained at different outdoor comprehensive temperatures by trying different proportions of the shading area, as illustrated in Figure 12.

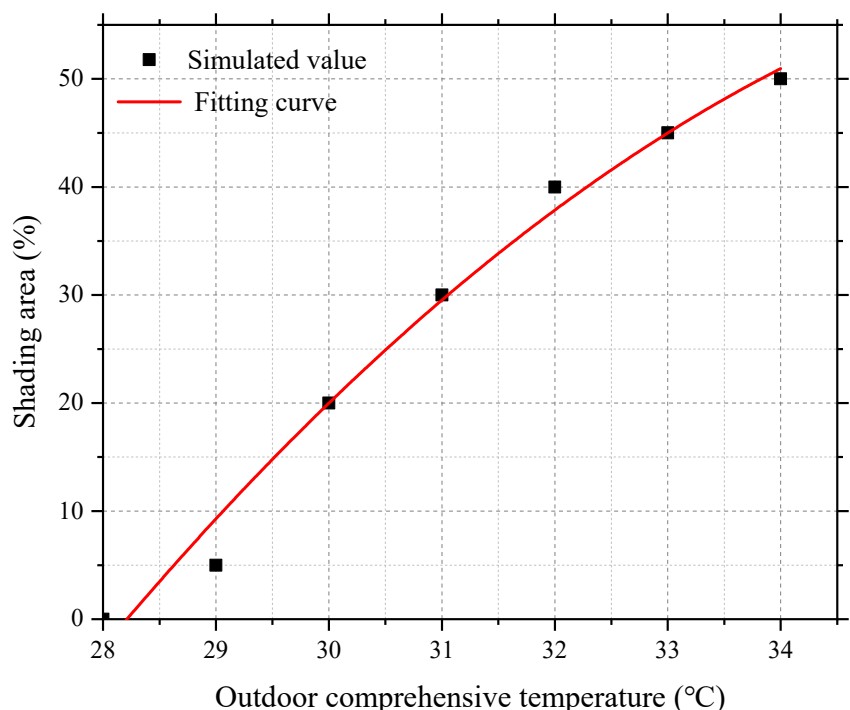

**Figure 12.** The fitting curve of outdoor comprehensive temperature and the required shading area.

When the outdoor comprehensive temperature ranges from 28 °C to 34 °C, and all the ventilation openings are kept open, the proportion of the shading area necessary to meet the indoor temperature requirements is obtained. The curve between the outdoor comprehensive temperature and the required shading area can be fitted with the equation of $y = -0.6x^2 + 45.8x - 819.3$, where x is the outdoor comprehensive temperature and y is the proportion of the required shading area. When the outdoor comprehensive temperature is lower than 28.6 °C, the greenhouse needs no shading to meet the indoor temperature requirements under full natural ventilation capacity. With the increase in outdoor comprehensive temperature, the growth rate of necessary shading area decreases. When the outdoor comprehensive temperature reaches 34 °C, the required shading area is close to 50%.

When the outdoor comprehensive temperature is 28.6 °C, the simulation results of the greenhouse temperature at 2 m high are shown in Figure 13. The average indoor temperature is 35.41 °C. For the height of 2 m, the temperature of 2/3 of the area is kept at about 30 °C, and only a small part of the area on the left has a higher temperature. The thermal environment meets the needs of plant survival and growth in general.

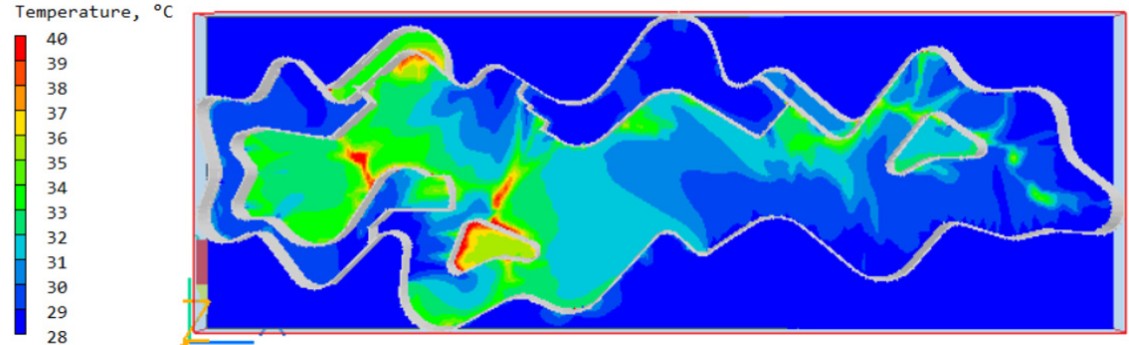

**Figure 13.** Contour of temperature distribution at 2 m high (outside, 28.6 °C).

In addition, when the outdoor comprehensive temperature is 34 °C, the cooling effect of different proportions of the shading area can be correlated with the fitting equation of $y = -9.1x^2 + 42.9$, as shown in Figure 14, where x is the proportion of shading area and y is the indoor temperature. As can be seen from Figure 14, when the outdoor comprehensive temperature is constant, the increase in the proportion of the shading area results in a steeper decrease in the indoor temperature, and shading can reduce the indoor temperature by about 3 °C at most.

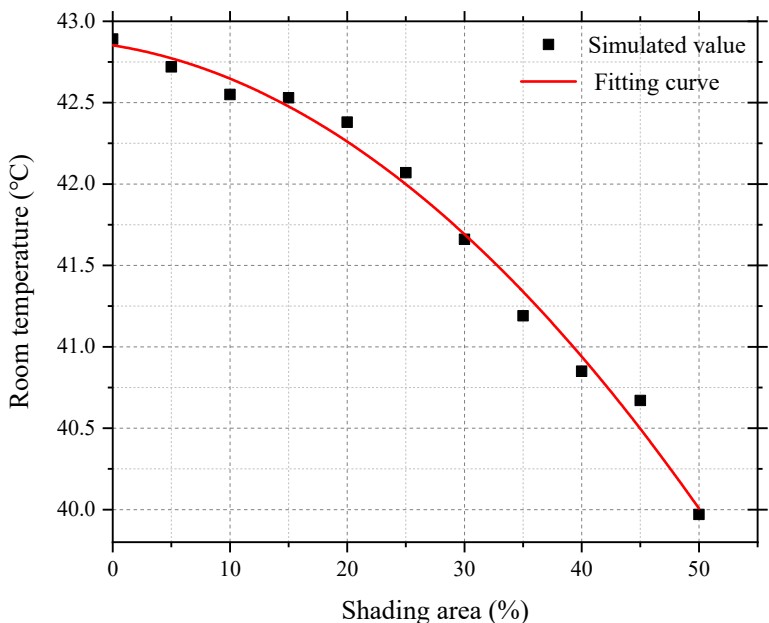

**Figure 14.** Fitting curves of different shading areas and indoor temperatures.

### 3.2. Analysis of Air Change Rate

Ventilation rate or air change rate is an important index to assess the indoor ventilation effect. "Greenhouse ventilation design code" [22] points out that from an economic point of view, the greenhouse air change rate should be less than 2 times/min, corresponding to 120 time/h. The indoor ventilation rate and the number of air changes are shown in Table 2, which shows that with the increasing outdoor comprehensive temperature and shading area, the natural ventilation rate gradually reduced. Under different operating conditions, the indoor ventilation rates do not vary significantly, and the values are generally 12–15 times/h.

**Table 2.** The number of air changes under different operating conditions.

| NO. | Outdoor Comprehensive Temperature | The Proportion of Shading Area | Ventilation Rate (m³/h) | Volume (m³) | Air Changes per Hour (Times/h) |
|---|---|---|---|---|---|
| 1 | 28 | 0% | $1.95 \times 10^6$ | | 14.6 |
| 2 | 29 | 5% | $1.93 \times 10^6$ | | 14.4 |
| 3 | 30 | 20% | $1.89 \times 10^6$ | | 14.2 |
| 4 | 31 | 30% | $1.73 \times 10^6$ | 133,560 | 13.0 |
| 5 | 32 | 40% | $1.67 \times 10^6$ | | 12.5 |
| 6 | 33 | 45% | $1.61 \times 10^6$ | | 12.1 |
| 7 | 34 | 50% | $1.55 \times 10^6$ | | 11.8 |

*3.3. Analysis of Greenhouse Carbon Emissions*

The issue of carbon emission is widely concerned with its publicly acknowledged climate change effect [23,24]. The greenhouse of the botanical garden not only performs as a carbon sink by plants consuming $CO_2$, but also turns into a carbon source by consuming cooling/heating energy to maintain the indoor temperature. As it adopts natural ventilation and utilizes shading cooling measures, compared with traditional cooling by air conditioning systems, it has the potential to greatly reduce carbon emissions [25].

The carbon emissions of buildings consist of the material production stage, transportation stage, operation and maintenance stage, dismantling stage, and recovery stage, of which the operation and maintenance stage can account for more than 85% [26–28]. Therefore, this paper does not consider the carbon emissions of other stages but focuses on the operation and maintenance of carbon emissions. The calculation equation for carbon emission in the operation and maintenance stage is

$$E_{YX} = \sum_{i=1}^{n}(AD_{YXDi} \cdot EF_{Di} \cdot l) \tag{6}$$

In Equation (6), $E_{YX}$ is carbon emission, $kgCO_2$; $AD_{YXDi}$ is electricity consumption, $kW \cdot h$; $EF_{Di}$ is grid emission factor, $EF_{Di} = 0.8095$ $kgCO_2$/ $kW \cdot h$; and $l$ is design life, a.

The annual hourly meteorological parameters of Shanghai can be obtained from typical meteorological year data. After screening, there are 604 h in the whole year when the outdoor comprehensive temperature is higher than 34 °C and cooling is required. The calculation formula of the cooling load formed by the solar radiation entering the room through the glass is

$$CL_W = C_{clW}C_zD_JF_W \tag{7}$$

In Equation (7), $CL_W$ is the hourly cooling load, W; $C_{clW}$ is the cooling load coefficient; $C_z$ is the shielding coefficient of glass; $D_J$ is the heat gain factor of solar radiation; and $F_W$ is the net area of glass, $m^2$.

The hourly cooling load calculation of heat transfer through glass curtain wall is

$$CL_E = KF(t_w - t_n) \tag{8}$$

In Equation (8), $CL_E$ is the hourly cooling load, W; K is the coefficient of heat transfer, $W/(m^2 \cdot K)$; F is the area of the glass curtain wall, $m^2$; and $t_w$ and $t_n$ are the calculated temperature of outdoor and indoor respectively, K.

According to the equations, the accumulative 604 h cooling load is 2,100,431 $kW \cdot h$. Assuming that the mechanical cooling system COP is 3.4, the annual cooling energy consumption is 617,773.82 $kW \cdot h$, corresponding to the annual carbon emission reduction of 500,087.91 kg $CO_2$. Therefore, by adopting natural ventilation and an exterior shading system, the botanical building saves around 500 t of $CO_2$ emissions every year.

## 4. Conclusions

In this paper, the thermal regulation of a greenhouse of a botanical garden in Shanghai is conducted, and the influence of shading area on indoor temperature is simulated by the CFD numerical method. It provides a reference for the control of shading to cool the greenhouse in summer under natural ventilation conditions:

(1)　The fitting equation $y = -0.6x^2 + 45.8x - 819.3$ was obtained between different outdoor comprehensive temperature and required shading area percentage. When the outdoor comprehensive temperature is below 28.6 °C, the greenhouse can be sufficiently cooled by natural ventilation without adopting shading. At the outdoor comprehensive temperature of 28.6 °C and above, with the increase in outdoor comprehensive temperature, the growth rate of the required shading area gradually decreases.

(2)　When the outdoor comprehensive temperature is 34 °C, the fitting equation between the indoor temperature and shading area is $y = -9.1x^2 + 42.9$, and shading can reduce the indoor temperature by about 3 °C.

(3)　When the designated indoor temperature is reached, outdoor comprehensive temperatures and shading areas will affect the natural ventilation rate. With the increase in outdoor comprehensive temperature and shading area, the indoor natural ventilation rate gradually decreases. In general, the air change rate is 12–15 times/h, sufficient to maintain the greenhouse thermal environment.

(4)　Compared with using air conditioning equipment to cool down the greenhouse, using natural ventilation assisted with shading can reduce the carbon emissions of the greenhouse at about 500 t $CO_2$ per year, and the energy saving and emission reduction effect are significant.

**Author Contributions:** Conceptualization, H.W.; methodology, H.W. and J.S.; software, J.S.; validation, J.W.; investigation, J.S. and J.W.; resources, J.W.; writing—original draft preparation, J.S.; writing—review and editing, H.W. and J.W.; visualization, J.S. and J.W.; supervision, H.W.; project administration, J.W.; funding acquisition, H.W. and J.W. All authors have read and agreed to the published version of the manuscript.

**Funding:** The APC was funded by the research project of the Shanghai Installation Group "Research and application of large space greenhouse airflow simulation and green control technology based on digital technology" under the project number 21JCSF-21.

**Institutional Review Board Statement:** Not applicable.

**Informed Consent Statement:** Not applicable.

**Data Availability Statement:** The data presented in this study are available on request from the corresponding author. The data are not publicly available due to the project owner's request.

**Conflicts of Interest:** The authors declare no conflict of interest.

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
