# Peer review of "CFD Simulation Study on the Cooling Characteristics of Shading and Natural Ventilation in Greenhouse of a Botanical Garden in Shanghai"

_sustainability, doi:10.3390/su15043056_

Round 1

Reviewer 1 Report

The paper determine the control strategy of shading area in the design and installation stage of a greenhouse.The paper provides some reference value for shading control and energy saving and emission reduction of greenhouse in botanical garden.

The description of this paper is clear and the logic is sound. The conclusions are good. I recommend to publish this work.

Author Response

Dear reviewers: thanks for the comments on our manuscript (The title is now revised per comments from one reviewer as “CFD simulation study on the cooling characteristics of shading and natural ventilation in greenhouse of a botanical garden in Shanghai”). Meanwhile, we have gone through the writing carefully to polish the writing and analysis. Please refer to the revised manuscript for details.

Reviewer 2 Report

A great summary of a huge work effort. 

Just to point out:

- Sometimes sentences are too long. It is better to use short and concise sentences to get the message across. Also, it is common to make the mistake of justifying and then asserting. 

- I have also seen the same term used in certain paragraphs. The same word is repeated several times. Paraphrasing is recommended.

Author Response

Dear reviewers: thanks for the comments to our manuscript (The title is now revised per comments from one reviewer as “CFD simulation study on the cooling characteristics of shading and natural ventilation in greenhouse of a botanical garden in Shanghai”). Per your comments, we have gone through the writing carefully to polish the writing and analysis. Please refer to the revised manuscript for details.

Reviewer 3 Report

The manuscript entitled "Study on the characteristics of shading and cooling in green-house of a botanical garden in Shanghai" investigated the influence of shading area on indoor temperature is studied by using CFD numerical method. The work is interesting but authors should rewritte the title and focus on CFD and results.  Moreover, the references in discussion are too few. Authors also should indicated the contributions of this study to field in the Introduction.

Author Response

Dear reviewers: thanks for the comments on our manuscript. Your comments are replied point by point as follows:

1. The manuscript entitled "Study on the characteristics of shading and cooling in green-house of a botanical garden in Shanghai" investigated the influence of shading area on indoor temperature is studied by using CFD numerical method. The work is interesting but authors should rewritte the title and focus on CFD and results. 

Reply: The title is now revised per comments as “CFD simulation study on the cooling characteristics of shading and natural ventilation in greenhouse of a botanical garden in Shanghai”).

2. Moreover, the references in discussion are too few. Authors also should indicated the contributions of this study to field in the Introduction.

Reply: We have added some references in the discussion and introduction part that highlight the contribution of this paper, please see the revised manuscript for details.

Reviewer 4 Report

The paper is related to CFD modeling of a particular configuration of the greenhouse. In the reviewer's opinion, some aspects of the model are missed in the paper, and the current version of the paper does not prove that the modeling is accurate enough.

In particular:

1) Equation 2-1 does not describe the mathematical model correctly (and there should be U instead u).

2) What are the boundary conditions used?

3) Are possible heat sources inside the greenhouse taken into account?

4) What is t_w in equation 2-5?

5) The mesh-dependence study seems to be insufficient. The conclusion on the low mesh sensitivity is based on a particular data set (Figure 7) that is not described enough.

6) 1700298.9 kgCO2 - seems to be too many digits in the number. What about the accuracy of the calculations?

7) The natural ventilation mechanism proposed is not described well. What kind of improvement is proposed? (opposite to the traditional air-conditioning equipment)

Round 2

Reviewer 4 Report

The paper is still far from the version to be published. The main issues are still related to the mathematical model description.

1) Equation (1) is still presented in the incorrect form. How does it correspond to the turbulent flow (momentum equation)? Why is this form of equation presented in the paper?

2) The window openings at the side wall and roof are treated as openings with given pressure. What were the inlet boundary conditions for the scalar quantities if the opening is a supply one?

3) What was the level of convergence (the level of residual decrease)?

4) What was the near-wall resolution (y-plus values)?

5) The 500087.91 kgCO2 has too many digits. Does the accuracy of simulation allow to evaluate the carbon emission reduction with so many digits? It is unbelievable. Approximate values are required.

Round 3

Reviewer 4 Report

The reviewer is satisfied with the answers. It seems that the current version of the paper satisfies the requirements of the journal and could be accepted.